# Transgenic quails reveal dynamic TCF/β-catenin signaling during avian embryonic development

Hila Barzilai-Tutsch[1,2], Valerie Morin[2], Gauthier Toulouse[2], Oleksandr Chernyavskiy[3], Stephen Firth[3], Christophe Marcelle[1,2]*, Olivier Serralbo[1]*

[1]Australian Regenerative Medicine Institute (ARMI), Monash University, Victoria, Australia; [2]Institut NeuroMyoGène (INMG), University Claude Bernard Lyon1, CNRS UMR, Lyon, France; [3]Monash Micro Imaging, Monash University, Clayton, Australia

**Abstract** The Wnt/β-catenin signaling pathway is highly conserved throughout evolution, playing crucial roles in several developmental and pathological processes. Wnt ligands can act at a considerable distance from their sources and it is therefore necessary to examine not only the Wnt-producing but also the Wnt-receiving cells and tissues to fully appreciate the many functions of this pathway. To monitor Wnt activity, multiple tools have been designed which consist of multimerized Wnt signaling response elements (TCF/LEF binding sites) driving the expression of fluorescent reporter proteins (e.g. GFP, RFP) or of LacZ. The high stability of those reporters leads to a considerable accumulation in cells activating the pathway, thereby making them easily detectable. However, this makes them unsuitable to follow temporal changes of the pathway's activity during dynamic biological events. Even though fluorescent transcriptional reporters can be destabilized to shorten their half-lives, this dramatically reduces signal intensities, particularly when applied in vivo. To alleviate these issues, we developed two transgenic quail lines in which high copy number (12× or 16×) of the TCF/LEF binding sites drive the expression of destabilized GFP variants. Translational enhancer sequences derived from viral mRNAs were used to increase signal intensity and specificity. This resulted in transgenic lines efficient for the characterization of TCF/β-catenin transcriptional dynamic activities during embryogenesis, including using in vivo imaging. Our analyses demonstrate the use of this transcriptional reporter to unveil novel aspects of Wnt signaling, thus opening new routes of investigation into the role of this pathway during amniote embryonic development.

**\*For correspondence:**
christophe.marcelle@univ-lyon1.fr (CM);
olivier.serralbo@csiro.au (OS)

**Competing interest:** The authors declare that no competing interests exist.

## Editor's evaluation

The manuscript describes several optimizations of classic DNA reporter constructs to monitor closely the dynamics of Wnt/β-catenin signalling during development using transgenic avian lines. As Wnt signalling pathway is essential in the homeostasis of vertebrate and invertebrate organisms, a robust tool to analyse finely the dynamics of the Wnt/β-catenin pathway is of broad interest to biology/biomedicine scientific communities.

## Introduction

The Wnt/β-catenin signaling pathway is a highly conserved pathway, which appeared early in phylogenesis and is common to all metazoan life forms. This pathway plays crucial roles during the entire lifespan of all organisms, from early embryogenesis to homeostasis in the adult. Wnt ligands bind to the transmembranal Frizzled receptor family and along with members of the LRP transmembranal

co-receptor family, they mediate a large array of cell responses including cell fate specification, polarization, migration, and mitogenic stimulation (*Clevers, 2006*; *Clevers and Nusse, 2012*; *Logan and Nusse, 2004*). The first and best characterized (and referred to as 'canonical') cellular response to Wnt is the inhibition of the β-catenin destruction complex, with the consequence of an increase of the β-catenin pool in the cytoplasm, ultimately leading to its translocation into the nucleus. There, it partners with members of the TCF/LEF family of transcription factors to activate various Wnt target genes, in a context-dependent manner.

To monitor the activity of the Wnt canonical pathway in vitro, transcription-based reporter systems were created by combining the DNA binding sites of TCF/LEF upstream of a minimal promoter and a reporter gene. The first reporter of Wnt/β-catenin signaling, TOPFlash, was used in vitro and it contained three TCF/LEF response elements upstream of a basal *c-fos* promoter driving the expression of the luciferase gene (*Korinek et al., 1997*; see also the Wnt Homepage for a complete list of references http://web.stanford.edu/group/nusselab/cgi-bin/wnt/). More sensitive TOPflash reporters were generated in *Drosophila* and zebrafish by increasing the number of TCF/LEF sites to 8, 12, and 16 (*DasGupta et al., 2005*; *Veeman et al., 2003*). In an attempt to detect Wnt signaling activity in mouse, three TCF/LEF binding sites were associated to LacZ and used to generate the TOPGAL mouse line (*DasGupta and Fuchs, 1999*), thus allowing the first analysis of Wnt responses in a vertebrate embryo. A significant increase in sensitivity was achieved by expanding the number of TCF/LEF binding sites to seven (BAT-gal mouse line; *Maretto et al., 2003*). Fluorescent reporter proteins (GFP or RFP and their variants) were also used in mouse and zebrafish (*Ferrer-Vaquer et al., 2010*; *Moro et al., 2012*). The main advantage of using either the β-galactosidase system or fluorescent proteins as reporters is their high stability: β-galactosidase half-life is reported to be up to 48 hr (*Egan et al., 2013*); that of GFP and RFP is about 24 hr (*Corish and Tyler-Smith, 1999*), and fusion of GFP to an H2B nuclear localization signal (as described in *Ferrer-Vaquer et al., 2010*) further stabilizes the fluorescent label (*Foudi et al., 2009*). Such high stabilities lead to a considerable accumulation of reporter proteins in cells activating the pathway, thus facilitating their detection. However, significant drawbacks are an important lag-time between the activation of the pathway and the detection of the reporter and, conversely, the detection of signals in tissues where Wnt activity may have already ceased. This makes stable reporters largely unsuitable to detect rapid spatiotemporal changes in a pathway activity. Destabilized fluorescent reporters have been designed to alleviate this problem; however, shortening their half-life leads to dramatic fluorescence signal losses: for instance, d2GFP (half-life 2 hr), is 90% less fluorescent than its native GFP counterpart (*He et al., 2019*). Combining four TCF/LEF binding sites with a destabilized fluorescent reporter (d2EGFP, 2 hr half-life, Clontech) in Zebrafish generated a transgenic line in which only intense activities of the pathway were detected through native fluorescence (*Dorsky et al., 2002*), thus requiring the more sensitive technique of in situ hybridization to detect lower Wnt signaling activities in this line. Increasing the number of TCF/LEF binding sites to six (upstream of a minimal promoter, miniP, and d2EGFP) generated a fish line with four insertion sites, in which many of the known Wnt/β-catenin signaling-active sites were detected by native fluorescence, including through live imaging (*Shimizu et al., 2012*).

Alternative reporter lines were also created by utilizing Wnt transcriptional targets (e.g. decrease 2 or LGR5) to generate transgenic or knock-in mouse lines (*Barker et al., 2007*; *de Roo et al., 2017*; *Lustig et al., 2002*; *Sonnen et al., 2018*; *van Amerongen et al., 2012*; *van de Moosdijk et al., 2020*). While those lines have been useful to characterize the targets' response to Wnt, they only partially cover all activities of the Wnt/β-catenin pathway.

Importantly, recent evidence suggests that mobilization of β-catenin from the cell membrane pool can also trigger the activation of TCF/LEF reporters in a WNT ligand-independent manner (*Lau et al., 2015*; *Sieiro et al., 2016*). This indicates that, even though TCF/LEF-based reporters faithfully reflect TCF/β-catenin transcriptional activity, it may not all be due to canonical Wnt signaling.

While strategies described above are mainly based on enhancing transcriptional activity of TCF/β-catenin reporters, very little has been done to reinforce their translational efficiency. Sequence elements in the 5' and 3' untranslated regions of mRNAs play crucial roles in translation and well characterized elements derived from plant and viruses have been successfully used in heterologous systems (cell culture and *Drosophila*) to considerably increase reporter protein yields (*He et al., 2019*; *Pfeiffer et al., 2012*).

Here, we generated two novel transgenic quail lines carrying TCF/LEF-responsive elements, using the technology we recently developed (*Serralbo et al., 2020*). In the first (named 12xTF-d2GFP), we have increased the number of TCF/LEF repeats to 12, upstream of a cytoplasmic, destabilized EGFP (d2EGFP). We had previously used this construct for electroporation of chicken embryonic tissues and shown that it is more sensitive to the activity of Wnt signaling than existing reporters containing three or eight TCF/LEF repeats (*Rios et al., 2010*). We had also shown that this accumulation of TCF/LEF repeats was not detrimental to the reporter accuracy, since the 12XTFd2EGFP responded positively to an activated form of β-catenin (*Rios et al., 2010*) and was strongly repressed by the Wnt-inhibiting molecule Dkk1 (*Sieiro et al., 2016*). Furthermore, we showed that the expression of the destabilized GFP reporter protein d2EGFP was similar to its mRNA, suggesting that 12XTFd2EGFP provides a precise view of cells actively engaged in Wnt signaling in vivo (*Rios et al., 2010*).

In the second line (named 16xTF-VNP), we increased transcriptional activity further, using 16 TCF/LEF repeats. This was combined with translational enhancers to drive the expression of a nuclear, destabilized, fast maturing Venus (*Nagai et al., 2002*; *Sonnen et al., 2018*) as reporter.

This resulted in two transgenic lines in which the characterization of the TCF/β-catenin transcriptional activity during embryogenesis is readily observed, including using in vivo imaging. Particularly remarkable is the 16xTF-VNP line, where intense, yet dynamic, reporter activity unveils unexpected features of TCF/β-catenin-responding cells and tissues.

This study underlines the importance of developing novel strategies to generate reporters efficient for monitoring of the spatiotemporal dynamics of signaling pathways and it opens new routes of investigation for correlating TCF/β-catenin transcriptional activities with unique cell behaviors during amniote embryonic development.

## Results and discussion
### Generation of the 12xTFd2GFP and 16xTF-VNP transgenic quail lines

To generate the 12xTF-d2GFP line (TgT2[12TCF/LEF:d2EGFP]), we modified a TCF/β-catenin transcriptional reporter we previously generated, which was intended for in vivo electroporation in chicken embryos (*Rios et al., 2010*; *Sieiro et al., 2016*) by inserting Tol2 (T2) transposable elements 5′ and 3′ of the construct, thus allowing its stable integration into the quail genome (*Figure 1A*). We used the direct injection technique as described in *Serralbo et al., 2020*; *Tyack et al., 2013* to transfect in vivo the blood-circulating primordial germ cells (PGCs). Fifty wild-type embryos at stage HH16 (E2.5) were injected in the dorsal aorta with a mix of lipofectamine 2000, the 12xTF-d2GFP plasmid and a pCAG-Transposase construct. Four founders were selected, of which one male was used as founder. It was mated with wild-type females and their embryos were used for the experiments. The transmission of the transgene to the offspring presented a Mendelian distribution, suggesting a single insertion.

To further improve the TCF/β-catenin reporter sensitivity, we generated the 16xTF-VNP line (TgT2[16TCF/LEF:Syn21-Venus-NLS-PEST-p10, Gga.CRYBB1:GFP]). For this, we synthesized and cloned 16 TCF/LEF repeats upstream of the TK minimal promoter, followed by IVS and Syn21 sequences (*Pfeiffer et al., 2012*), directly abutting the ATG initiation codon of Venus (*Figure 1B*). The IVS (Intervening Sequence) is a 67 bp long sequence from *Drosophila* myosin heavy chain, which facilitates mRNA export to the cytoplasm (*Pfeiffer et al., 2012*). Syn21 is an AT-rich 43 bp consensus translation initiation sequence made of elements derived from *Drosophila,* and from the *Malacosoma neustria* nucleopolyhedrovirus polyhedrin gene. We chose a nuclear, destabilized form of the EYFP variant Venus as reporter (1.8 hr half-life; *Abranches et al., 2013*), as it displays a 156% increase in relative brightness compared to EGFP (*Nagoshi et al., 2004*). It was followed by the p10 sequence, which is a 606 bp terminator sequence from the *Autographa californica* nuclear polyhedrosis baculovirus (*Pfeiffer et al., 2012*). A CrystallGFP selection mini-gene, consisting of the promoter of the βB1crystallin gene (active exclusively in the lens) upstream of EGFP was added to the construct to ease the selection of transgenic birds at hatching (*Serralbo et al., 2020*). Finally, Tol2 sites were added for stable integration into the quail genome.

To validate the efficiency of this construct, we electroporated one-half of the neural tube of HH15 (E2.5) chicken embryos with a DNA mix containing the 16xTF-VNP construct and a TagBFP protein driven by the CAG ubiquitous promoter as a marker of electroporated cells. Twenty-four hours after electroporation, weak expression of 16xTF-VNP was observed in the dorsal neural tube (NT) of the

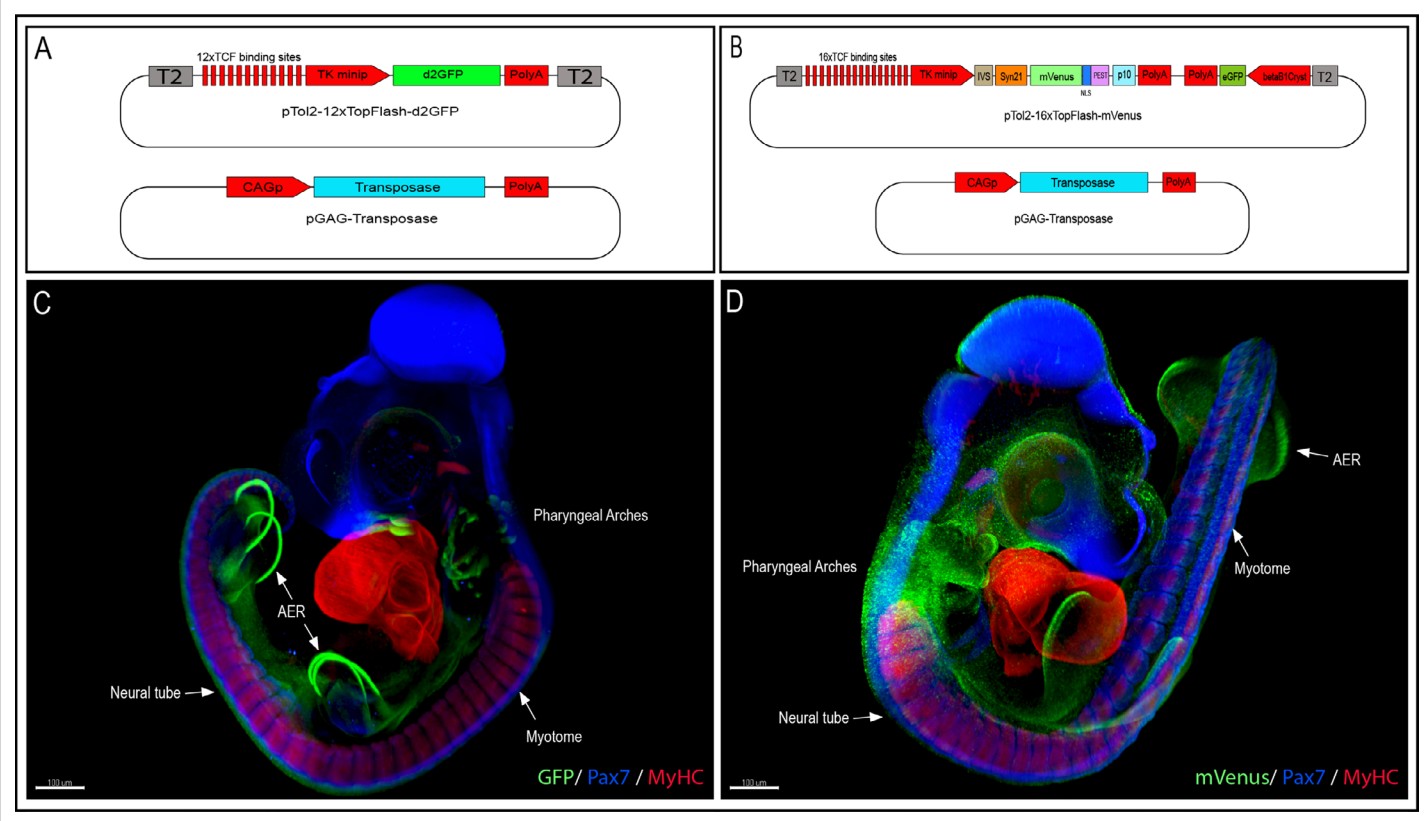

**Figure 1.** Generation of the 12xTF-d2GFP and the 16xTF-VNP transgenic Japanese quail lines. (**A**) Vectors used to generate the 12xTFd2GFP line and (**B**) the 16xTF-VNP line. (**C**) An E3.5 12xTFd2GFP embryo and (**D**) an E3 16xTF-VNP embryo cleared with the 3DISCO method, showing an overview of the TCF/β-catenin reporter activities in these lines. Embryos stained for Pax7 (blue), MyHC (red), and GFP or mVenus (green). Scale bar 100 µm. AER, apical ectodermal ridge.

The online version of this article includes the following source data and figure supplement(s) for figure 1:

**Figure supplement 1.** Expression of the 16xTF-VNP in the neural tube and migrating neural crest.

**Figure supplement 1—source data 1.** Source data file for the quantification shown in *Figure 1—figure supplement 1*, panel N.

**Figure supplement 2.** Comparison of reporter fluorescence levels.

**Figure supplement 2—source data 1.** Source data file for the quantification shown in *Figure 1—figure supplement 2*, panel J.

**Figure supplement 3.** TCF/β-catenin reporter expression in E4 embryos.

electroporated half, while a stronger expression of the reporter was seen in the migrating neural crest cell population (NC; *Figure 1—figure supplement 1A-D*). The co-electroporation of the CAG-TagBFP as internal control and the 16xTF-VNP reporter, with or without a dominant-negative form of LEF1 (DN-LEF1; *Figure 1—figure supplement 1H-J*), or the Wnt-inhibiting molecule Axin2 (*Figure 1—figure supplement 1K-M*), led to a 9- and 3.8-fold decrease, respectively, in the number of cells expressing the 16xTF-VNP construct out of the total BFP-positive cells (*Figure 1—figure supplement 1N*), as opposed to control embryos. These results suggest that the accumulation of 16 TCF/LEF repeats is not detrimental to the reporter accuracy.

To test the added value of translation enhancers in a vertebrate environment, we compared the fluorescence levels obtained with the 16xTF-VNP, the 12xTFd2GFP, and a 16xTF-VNP construct lacking the translation enhancers (referred to here as 16xTF-VNP plain). We co-electroporated the right side of the neural tube of E2.5 chicken embryos with the 12xTFd2GFP, 16xTF-VNP plain, or the 16xTF-VNP, together with CAG-TagBFP as internal control (*Figure 1—figure supplement 2*). Twenty-four hours post-electroporation, we imaged the native expression of the fluorescent reporters and calculated the intensity of fluorescence (pixel number and intensity) in the green channel (GFP or Venus) normalized to the intensity of fluorescence in the blue channel (BFP). We observed an 11-fold increase in the fluorescence level of 16xTF-VNP plain reporter compared to 12xTFd2GFP reporter, and a 1.6-fold

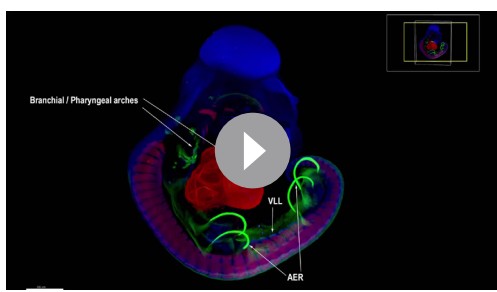

**Video 1.** A 3D reconstruction of E3.5 12xTFd2GFP embryo cleared with the 3DISCO method. The embryo is stained for GFP (green), Pax7 (blue), and MyHC (red). https://elifesciences.org/articles/72098/figures#video1

increase in the fluorescence level of the 16xTF-VNP compared to the 16xTF-VNP plain reporter (*Figure 1—figure supplement 2J*). Altogether, this suggests that the 16xTF-VNP construct is a reliable and sensitive reporter of TCF/β-catenin transcriptional activity, suitable to monitor the dynamics of its transcriptional activity.

We used the same technique as above to generate transgenic quails with this reporter. One female was selected as the transgenic founder using the CrystalIGFP marker and crossed with a WT male to expand the transgenic line. The transmission of the transgene to the offspring presented a Mendelian distribution, suggesting a single insertion.

## TCF/β-catenin transcriptional activities in early embryos

To characterize TCF/β-catenin transcriptional activity during embryonic development, we analyzed and compared the activity of the two reporters in the two lines we generated, in whole mount preparations of immunostained embryos, on immunostained sections and in live tissues at different developmental stages. Whole mount preparations of HH21 (E3.5; 12xTF-d2GFP) and HH20 (E3; 16xTF-VNP) transgenic quail embryos immunostained for GFP or mVenus, Pax7 and Myosin Heavy Chain (MyHC), clarified by the '3DISCO' technique (*Belle et al., 2017*) and imaged using LaVision BioTec UltraMicroscope II, provide an overview of the sites of high reporter activities. It shows conspicuous reporter activity in the apical ectodermal ridge (AER), mesenchymal limb bud cells, the pharyngeal arches (particularly the maxillary), the somites, and migrating neural crest (*Figure 1C, D*; *Videos 1 and 2*). This initial examination also indicated that the 16xTF-VNP reporter line is more sensitive to TCF/β-catenin transcriptional activity than the 12xTF-d2GFP, as it shows reporter activity in places that are not readily detected in the 12xTF-d2GFP line (e.g. the cephalic neural crest; *Figure 1D*). This observation was seen also in HH23 (E4) 12xTFd2GFP and 16xTF-VNP embryos immunostained for GFP or mVenus, Neurofilament and MyHC, clarified by the '3DISCO' technique (*Belle et al., 2017*) and imaged as mentioned above (*Figure 1—figure supplement 3*). While such technique is attractive, as it gives a general overview of the reporter's activity in all tissues within a developing embryo, it lacks the sensitivity that can be obtained with more classical approaches, such as immunohistochemistry on sections. We therefore performed transverse sections of E3 12xTF-d2GFP and 16xTF-VNP embryos stained for GFP, Pax7, and MyHC which confirmed that TCF/β-catenin transcriptional activity was generally more prominent in the 16xTF-VNP than in the 12xTF-d2GFP (*Figure 2*). For instance, reporter activity was observed throughout the dermomyotome (stronger in the medial and lateral border, DML and VLL, respectively) in the 16xTF-VNP, while it was visible only in the DML and VLL in the 12xTF-d2GFP (*Figure 2A–D*). In both reporter lines, significant reporter expression was present in mesenchymal cells within the limb bud, and strong activity was observed in the AER (*Figure 2A, C*).

We made surprising observations in the neural tube and neural crest (NC). We observed that the activity of the reporter was not detected in the dorsal neural tube in the 12xTF-d2GFP (*Figure 2A, B* and *Figure 2—figure supplement 1A-J*) and weakly active in the more sensitive 16xTF-VNP line (*Figure 2C, D* and *Figure 2—figure supplement 1K-T*). In contrast, migrating NC strongly upregulated the reporter as they left the neural tube *en route* to their sites of differentiation (this is particularly visible in the 16xTF-VNP; *Figure 2C, D* and *Figure 2—figure supplement 2A-D*).

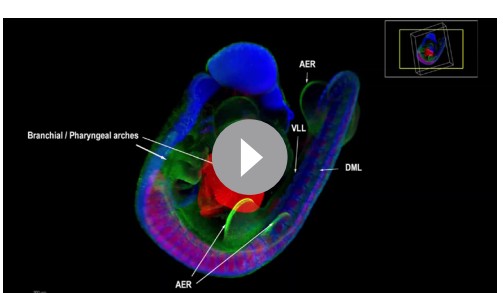

**Video 2.** A 3D reconstruction of E3 16xTF-VNP embryo cleared with the 3DISCO method. The embryo is stained for GFP (green), Pax7 (blue), and MyHC (red). https://elifesciences.org/articles/72098/figures#video2

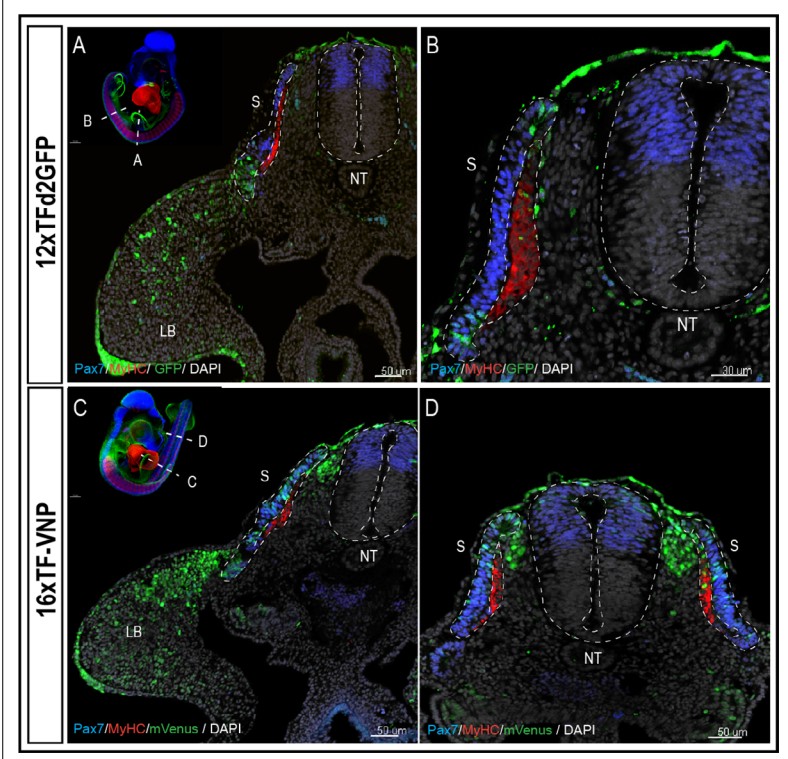

**Figure 2.** The 16xTF-VNP reporter is a sensitive reporter of the TCF/β-catenin signaling activity. Transverse sections at the levels of the front limb (**A, C**) and the trunk (**B,D**) of E3 12xTF-d2GFP (**A–B**) and 16xTF-VNP (**C–D**) transgenic embryos stained for GFP or mVenus (green), Pax7 (blue), MyHC (red), and DAPI (gray). Inserts show the levels at which the sections were made. Scale bar 50 μm (**A,C,D**) or 30 μm (**B**). S, somite; NT, neural tube; LB, limb bud.

The online version of this article includes the following figure supplement(s) for figure 2:

**Figure supplement 1.** Transverse sections at the levels of the front limb and the trunk of E3 12xTFd2GFP and 16xTF-VNP transgenic embryos stained for GFP or mVenus, respectively (green), Pax7 (blue), MyHC (red), and DAPI (gray).

**Figure supplement 2.** TCF/β-catenin reporter dynamic activity in NC cells.

**Figure supplement 3.** In situ hybridization of Wnt3a and Wnt1 mRNA in quail embryos.

These patterns of the reporters' activities are unexpected, regarding the published expression patterns of Wnt mRNAs and proteins in those tissues. Previous works have shown that Wnt1 and Wnt3a mRNA transcripts are present in the roof plate (RP) of the neural tube in mouse and chicken early embryos while their transcripts are absent from migrating NC (*Hollyday et al., 1995*; *Marcelle et al., 1997*). Wnt1 and Wnt3a mRNA transcripts are still present in the RP of quail embryos at the time of our analyses (*Figure 2—figure supplement 3A-D*). Remarkably, the Wnt1 protein (Wnt3a was not tested), is loaded onto NC, as they initiate their migration, to be delivered at a distance to somites where it serves to regulate myotome organization (*Serralbo and Marcelle, 2014*). The observation we made here brings another level of understanding of the roles of Wnt in those tissues, since it indicates that at the time of observation, epithelial cells of the dorsal neural tube and the RP poorly respond to Wnt, while the epithelial-mesenchyme transition, which is necessary for NC migration, triggers a strong increase in the reporter response in those cells. As NC proceeded along their dorso-ventral migration path, the reporter activity rapidly diminished, generating a gradient of TCF/β-catenin transcriptional activity likely due to the exhaustion of the pool of Wnt ligand initially loaded on the NC cell surface (*Figure 2C, D* and *Figure 2—figure supplement 2A-D*). This indicates that NC not only deliver Wnt at a distance to somites, but they also temporally use it for their own purpose. The functions of Wnt signaling in the CNS and the NC have been extensively investigated. Loss and gain-of-function of Wnt1 and/or 3a in mouse, *Xenopus* and chicken have led to the premise that these

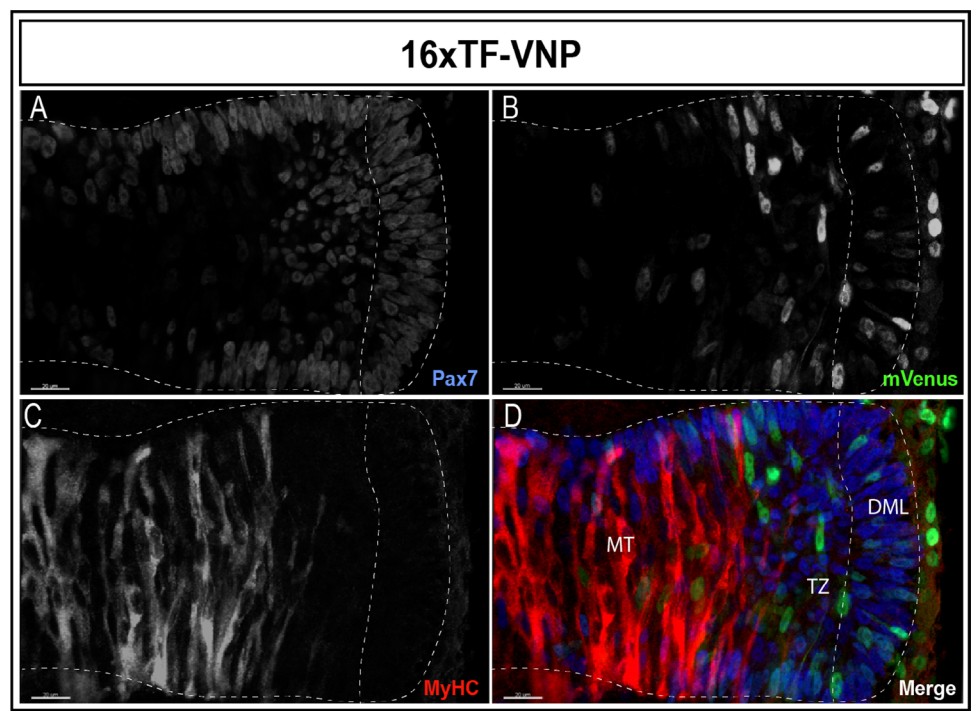

**Figure 3.** Dynamic TCF/β-catenin signaling activity in somites. An optical section of an E2.5 embryo somite stained for Pax7 (**A,D**); blue mVenus (**B,D**); green, and MyHC (**C,D**); red reveals high and low reporter expression levels in epithelial cells of the DML, while high expression is seen in elongating myocytes located in the transition zone (TZ). The reporter levels are decreasing in fully elongated myocytes of the myotome. Scale bar 20 μm. DML, dorso-medial lip; MT, myotome.

molecules play important functions in the proliferation and/or differentiation of neuronal precursors within the CNS, opposing a Shh gradient from the notochord and floor plate (*Alvarez-Medina et al., 2008*; *Dessaud et al., 2008*; *Dickinson et al., 1994*; *Ikeya et al., 1997*; *McMahon et al., 1992*; *Saint-Jeannet et al., 1997*). However, patterning defects in Wnt1 or compound Wnt1/Wnt 3a mutants were only observed in the brain, while the spinal cord was unaffected (*Ikeya et al., 1997*; *McMahon et al., 1992*). In neural crest, loss of Wnt-1 and Wnt-3a functions had a major impact, leading to a large deficit in all their derivatives (e.g. in cranial and dorsal root ganglia) that suggested a role in the neural crest cell proliferation (*Ikeya et al., 1997*). Wnt signaling may also play a role in NC differentiation; however, this function is unclear, since its inhibition or the activation in neural crest cells were reported to promote neuronal fates at the expense of other derivatives (*Dorsky et al., 1998*; *Lee et al., 2004*). The availability of sensitive and dynamic tools to monitor TCF/β-catenin transcriptional activity sheds new light on the time window and the place when Wnt signaling is likely to be active in the CNS and in the changes observed in the transcriptional signature of neural crest cells along their migratory routes (*Azambuja and Simoes-Costa, 2021*; *Morrison et al., 2017*; *Simões-Costa and Bronner, 2015*).

A similar gradient of reporter activity was observed in somites, since it was active in the DML and the transition zone and rapidly decreased in the myotome (*Figure 3A–D*). As previously described, high and low TCF/β-catenin transcriptional activities were observed within DML cells, the strongly labeled ones corresponding to DML cells engaging into the myogenic program (see also below; *Sieiro et al., 2016*).

Thus, the use of destabilized fluorescent reporters to generate these quail lines allows the detection of dynamic changes (increase and decrease) in reporter activity throughout embryonic development that could not be appreciated with stable reporters, such as the BAT-gal mouse line (*Maretto et al., 2003*).

## TCF/β-catenin activity during early embryonic development

To characterize TCF/β-catenin activity during early embryonic development, we performed immunostaining on sections and on whole mount embryos, coupled with classical confocal microscopy. We observed that the reporters were first detected at the posterior end of a gastrulating embryo (stage HH4, about E1) in the 16xTF-VNP embryos (*Figure 4—figure supplement 1*) and two stages later (HH6) for the 12xTFd2GFP (data not shown). We then examined embryos at developmental stage HH12 (E2, *Figure 4* and *Figure 4—figure supplement 2*). It is visible that the reporter is expressed similarly in both lines but more conspicuously in the 16xTF-VNP line. For instance, the fluorescent signal is barely visible in the tailbud of the 12xTFd2GFP while it extends anteriorly along one-third of the presomitic mesoderm (PSM) region in the 16xTF-VNP line. Similarly, migrating cephalic neural crest cells (e.g. around the otic vesicle) were more prominently labeled in the 16xTF-VNP line than in the 12xTFd2GFP. These observations further illustrate the improvement brought about by the 16xTF-VNP construct over the 12xTFd2GFP construct. The similarity of the reporter expression in both lines also supports the premise that there is little if any positional effect due to the insertion sites of the transgenes.

To gain a cellular resolution of the 16xTF-VNP reporter activity, we prepared transverse sections of HH14 (E2.5) embryos at the level of cervical somites, somite I (the newly formed somite) and the PSM, and stained for mVenus, Acetylated Tubulin and DAPI (*Figure 5*). These sections show the nuclear localization of the VNP, and further demonstrate the dynamic changes of the reporter in the developing embryo. At the PSM level (*Figure 5I-L*), the reporter was strongly expressed in ectodermal cells, and in the entire NT. Few cells were positive for VNP in the segmental plate mesoderm. In somite I (*Figure 5E-H*) conspicuous expression of the reporter was observed in the dorsal NT, while its activity was absent in the ventral part of the NT. A salt-and-pepper expression was observed in the dorsal part of somite I with high and low levels of expression in individual cells. The reporter was active in ectodermal cells, particularly dorsal to the lateral plate mesoderm cells. In the cervical somite level (*Figure 5A-D*), the reporter expression pattern was similar to that in somite I. However, its activity in the dorsal NT was reduced at a time when the first, strongly labeled neural crest cells emanate from the neural tube.

## TCF/β-catenin activity during late organogenesis

We also analyzed the activity of the reporters at 9 days of development (HH35; *Figure 6*). Tissues that displayed strong TCF/β-catenin transcriptional activity include the egg tooth (*Figure 6A, F*), the liver (*Figure 6B, F*), the feather buds (*Figure 6C*), the embryonic vertebrae (*Figure 6D*), as well as the limb bone growth zones (*Figure 6E*). While the significance of the conspicuous reporter activity in egg tooth formation is unknown, Wnt functions in liver, feather follicles and bone formation during organogenesis are well documented. Wnt/β-catenin was shown to promote hepatocyte proliferation in mice (*Perugorria et al., 2019*), to initiate the formation of hair and feather follicle placodes in mouse and chicken (*DasGupta and Fuchs, 1999*; *Fuchs, 2016*; *Noramly et al., 1999*; *Olivera-Martinez et al., 2001*) and to favor osteoblast differentiation over chondrocyte differentiation, thereby determining whether mesenchymal progenitors become osteoblasts or chondrocytes (*Baron et al., 2006*; *Day et al., 2005*).

## Dynamic TCF/β-catenin transcriptional activity in live tissues

TCF/β-catenin spatiotemporal activity is highly dynamic, both on a cellular level and throughout development, a feature that has been difficult to study due to lack of reliable destabilized reporters. By using the 12xTF-d2GFP and the 16xTF-VNP transgenic lines, we followed the dynamics of TCF/β-catenin transcriptional activity in vivo. Observation of early somites showed a strong activation of the 12xTF-d2GFP reporter activity in single cells located in the DML (*Video 3*, arrowheads; see also *Video 3 – Figure 4—figure supplement 1*). The timeline of this movie shows that the increase of signal (from lowest to highest) takes place over a period of 5–6 hr. This is coherent with our previous studies, which showed that single epithelial cells within the DML, receiving Delta signals from incoming migrating neural crest cells, respond by activating the myogenic program through a NOTCH/β-catenin-dependent/Wnt-independent signaling module (*Rios et al., 2011*; *Sieiro et al., 2016*). The in vivo analysis presented here suggests that the entry of DML cells into myogenesis can be monitored

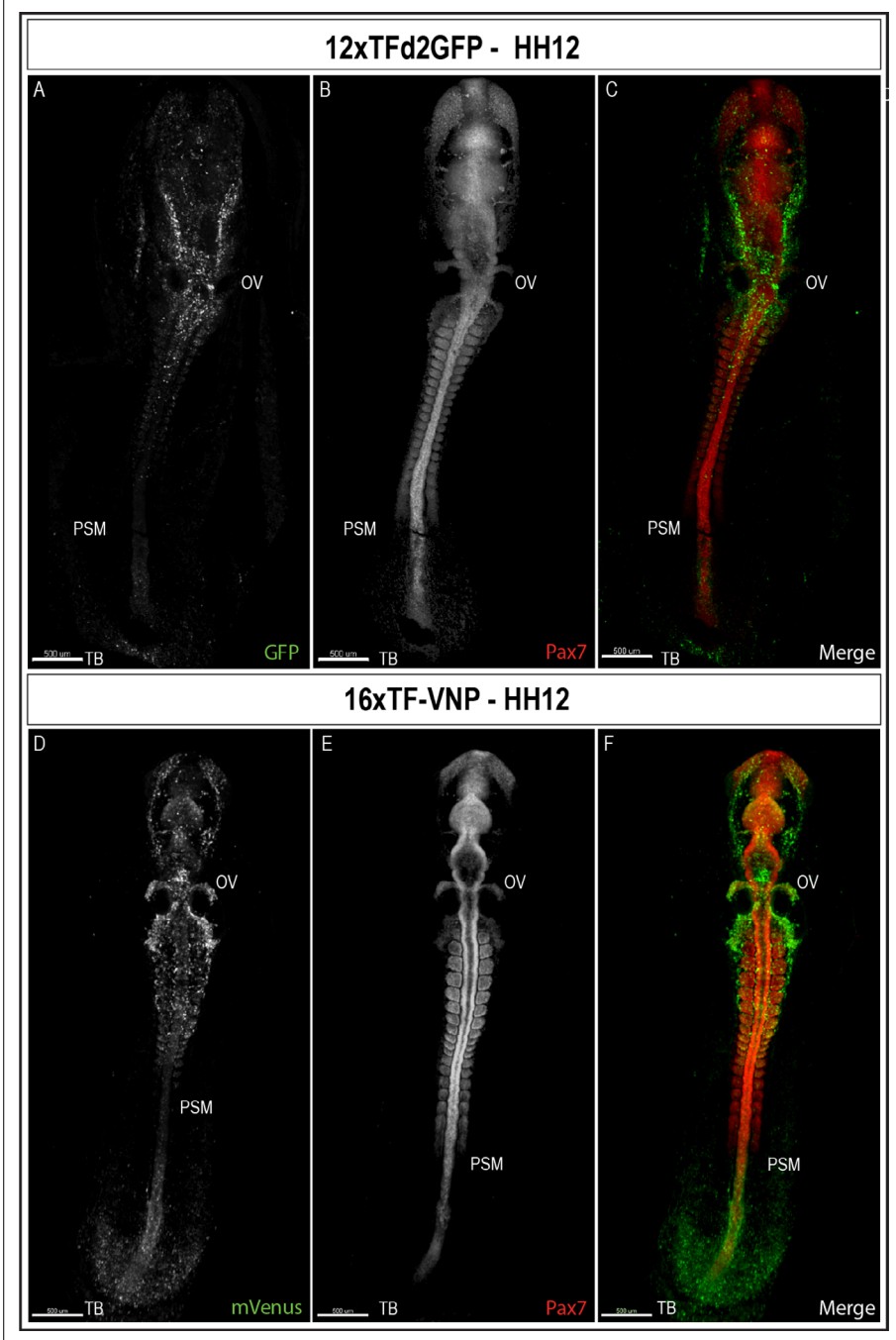

**Figure 4.** TCF/β-catenin reporter expression in HH12 embryos. Whole-mount view of HH12 12xTFd2GFP (**A–C**) and 16xTF-VNP (**D–F**) embryos immunostained for GFP or mVenus (green) and Pax7 (Red). Both embryos present strong TCF/β-catenin reporter activity in migrating cephalic neural crest in the head area, somites, the posterior neural tube and the tail bud area. Scale bar 500 μm. OV, otic vesicle; PSM, presomitic mesoderm; TB, tail bud.

The online version of this article includes the following figure supplement(s) for figure 4:

**Figure supplement 1.** Whole-mount view of HH4 16xTF-VNP embryo immunostained for mVenus (green), Pax7 (red), and DAPI (blue).

**Figure supplement 2.** Whole-mount view of HH11 16xTF-VNP embryo immunostained for mVenus (green) and Pax7 (red).

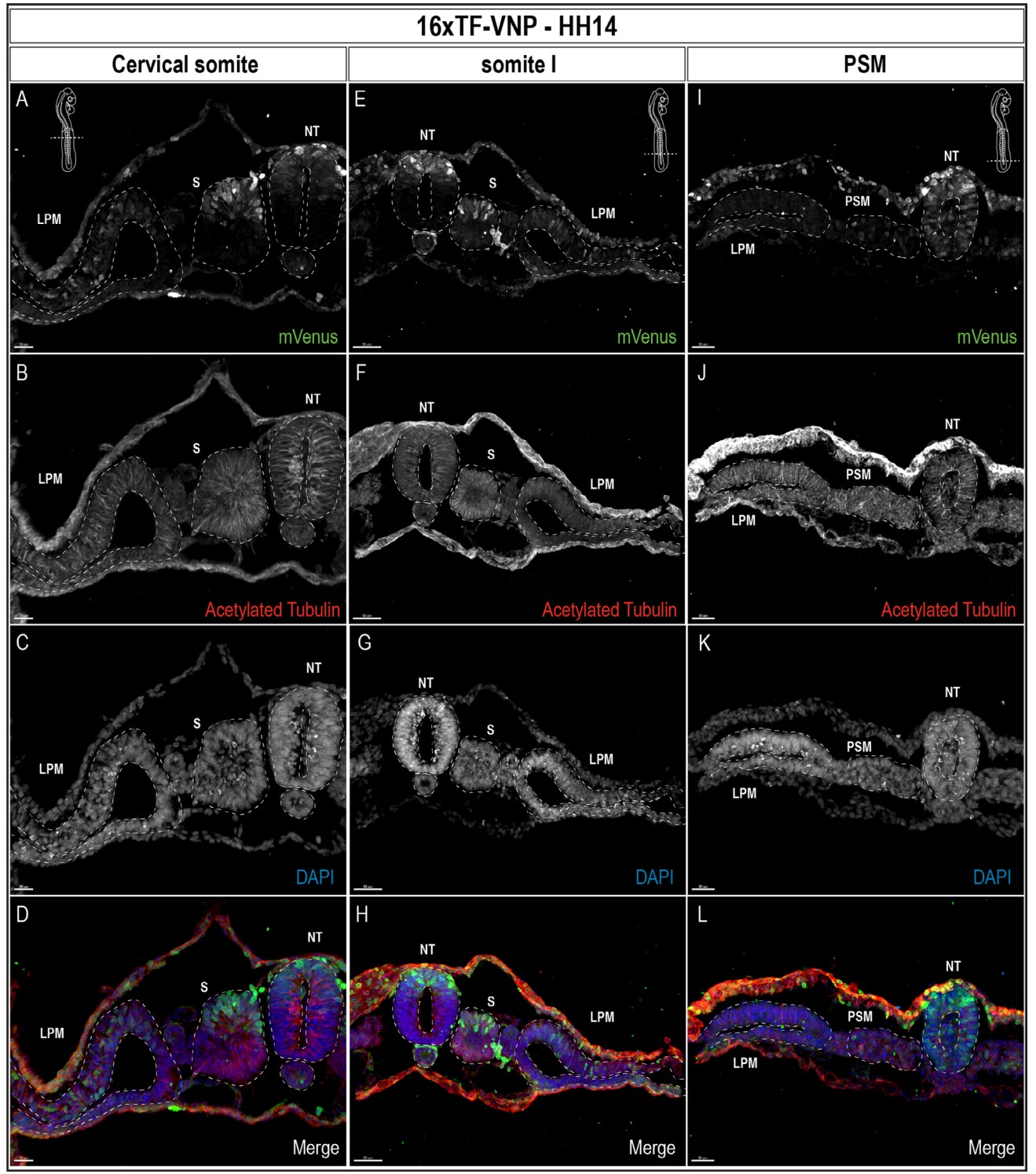

**Figure 5.** TCF/β-catenin reporter expression pattern in transverse sections of a 14HH 16xTF-VNP embryo. Transverse sections at the level of the cervical somites (**A–D**), somite I (**E–H**), and the anterior PSM (**I–L**) of a 14HH 16xTF-VNP embryo, immunostained for mVenus (green), Acetylated Tubulin (red), and DAPI (blue). At the PSM level, Venus is expressed in the entire neural tube, in individual ectodermal cells and at low level in the PSM. At somite I level, mVenus is observed in the dorsal neural tube, the dorsal part of the somite, the ectoderm and the lateral plate mesoderm. At the level of cervical

*Figure 5 continued on next page*

*Figure 5 continued*

somites, mVenus expression is faint in the dorsal neural tube and it is stronger in migrating neural crest cells. The nuclear localization of mVenus is detectable by its colocalization with DAPI. Inserts show the levels at which the sections were made. Scale bar 20 μm (**A–D**), 50 m μm (**E–H**), or 30 μm (**I–L**). NT, neural tube; S, somite; LPM, lateral plate mesoderm.

through an increase in TCF/β-catenin reporter activity and their behavior followed live using the quail lines we generated.

A possible limitation of the TCF/β-catenin reporters we designed resides in the half-life of the chromophores we used (i.e. d2EGFP and VNP), which has been estimated to be about 2 hr (*Li et al., 1998*; *Abranches et al., 2013*). This suggests that dynamic changes in Wnt signaling close to, or below, 2 hr will be difficult to detect with these tools.

An interesting finding of this study was the intense TCF/β-catenin-response in the fore- and hindlimb AER. The AER remained strongly labeled throughout limb growth (up to E9, where it was still detected at the fingertips, *Figure 6E*). Interestingly, the reporter-positive cells were found scattered in a wide region of early limb ectoderm (*Video 4* and *Figure 5* right panel), intermingled

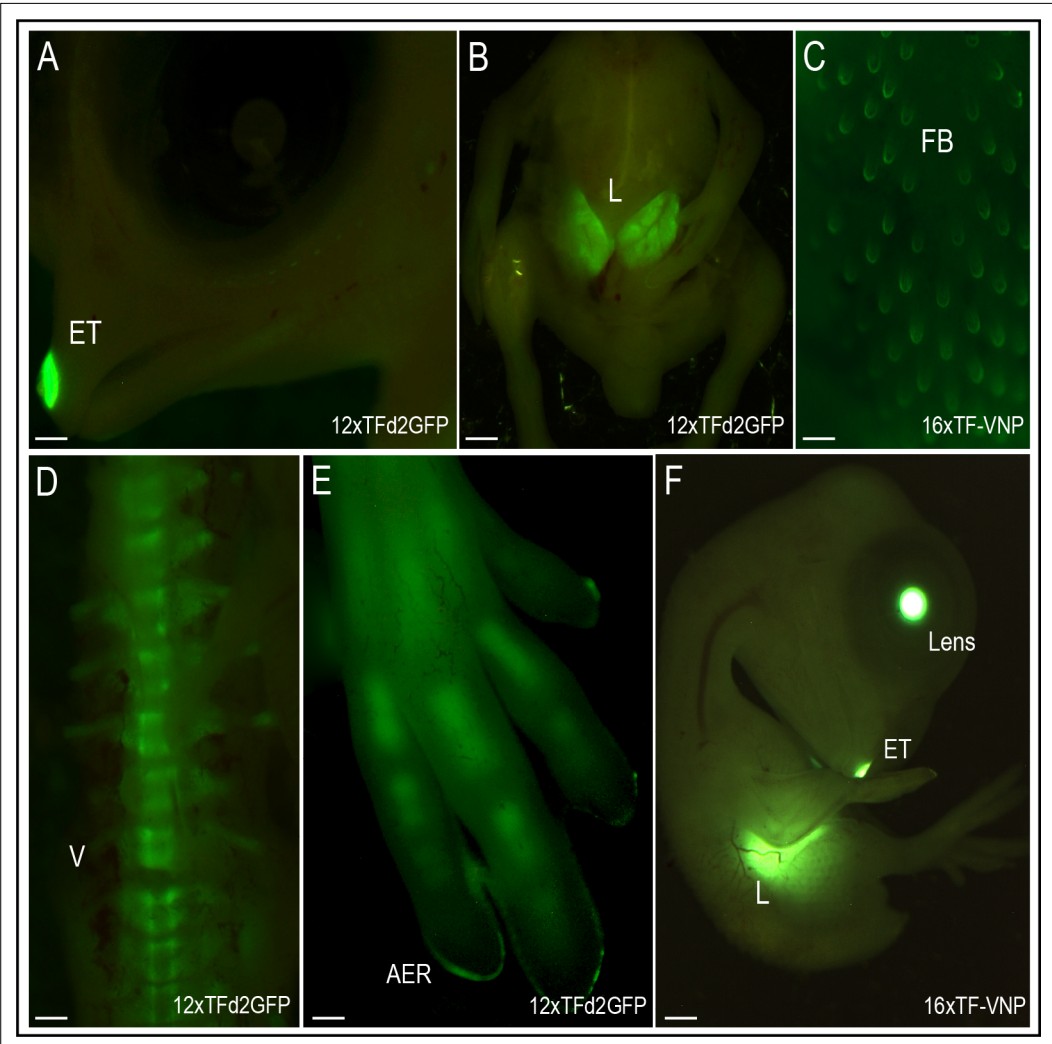

**Figure 6.** TCF/β-catenin reporter activity in late developmental stages. Native TCF/β-catenin signaling reporter activity in E9 12xTF-d2GFP (panels **A,B,D,E**) and E8 16xTF-VNP embryos (**C,F**). In the 12xTFd2GFP embryo the TCF/β-catenin reporter is strongly expressed in the egg tooth (ET, panel **A**), liver (L; panel **B**), AER (panel **E**) and differentiating bones (vertebrae, V, panel **D**) and digits, panel (**E**). In the 16xTF-VNP embryo, the reporter activity is also observed in the feather follicles (FB; panel **C**), the egg tooth and the liver (panel **F**). The transgenic embryo Crystallin-EGFP marker is also visible in the lens of the 16xTF-VNP embryo. Scale bar 300μm.

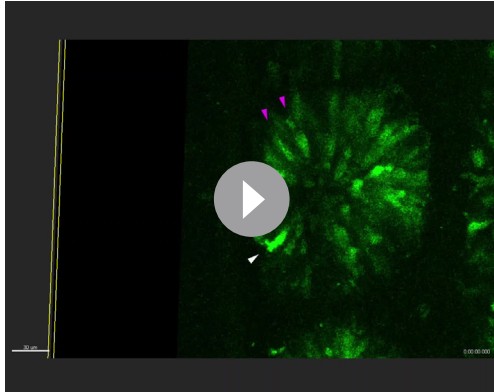

**Video 3.** A 9 h time-lapse confocal analysis of an E2.5 12xTFd2GFP embryo somite. The movie shows a single Z-plane of the somite (10 μm) and focuses on two cells in the dorsal DML which increase the TCF/β-catenin reporter activity (magenta arrowheads). We also show a cell division event in the caudal DML (white arrowheads).

https://elifesciences.org/articles/72098/figures#video3

with reporter-negative cells. As the limbs grew, reporter-positive cells migrated toward and coalesced to form the AER, suggesting that they constitute a population of AER progenitors. The AER is crucial to limb formation, which serves as a signaling center that regulates dorso-ventral patterning of the limb and its proximo-distal growth (*Fernandez-Teran and Ros, 2008*; *Zeller et al., 2009*). Interestingly, lineage analyses performed in chicken embryos had shown that the AER is derived from cells located on a wide region of early ectoderm (*Altabef et al., 1997*; *Michaud et al., 1997*) and had suggested that AER progenitors were intermingled with non-ridge progenitor, an observation coherent with our finding. Wnt3a in the chicken embryo follows a similar pattern to that of the reporter, widely expressed throughout the dorsal ectoderm in early developing limb buds, and later condensing to the AER region (*Fernandez-Teran and Ros, 2008*; *Kengaku et al., 1998*; *Zeller et al., 2009*). While it is possible that AER progenitors (recognized by their expression of the TCF/β-catenin reporter) are specified and/or respond to the Wnt3a signal, a recent study suggested that mechanical tensions from the underlying growing limb mesenchyme participate in the activation of TCF/β-catenin response (and therefore in the specification of AER progenitors) in the overlying ectoderm in a Wnt-independent manner (*Lau et al., 2015*). In this context, it will be interesting to determine whether Wnt3a acts as a directional cue in the migration of AER progenitors toward the AER anlage.

In summary, we have generated two transgenic quail lines suitable for the study of the dynamic behavior of TCF/β-catenin signaling, with the 16xTF-VNP being the most sensitive. Both lines allow to overcome previous limitations in the study of the Wnt/TCF/β-catenin signaling, particularly in vivo and their availability opens new routes of investigation into dynamic signaling activity of this pathway throughout development.

# Materials and methods

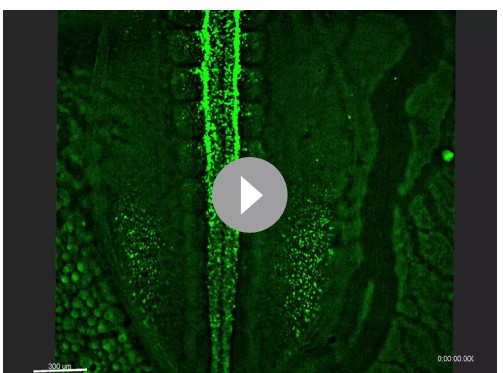

**Video 4.** A 9 hr time-lapse movie of an E2 16xTF-VNP embryo showing the growing hind limb bud. Ectodermal cells, strongly expressing the TCF/β-catenin reporter, are seen as they migrate toward the AER region where they condensate. The migrating NC and somitic cells are also highly visible.

https://elifesciences.org/articles/72098/figures#video4

## Generating transgenic quail by direct injection

The direct injection technique was performed as described in *Serralbo et al., 2020*; *Tyack et al., 2013*. The injection mix contained 0.6 μg of Tol2 plasmid, 1.2 μg of CAG-Transposase plasmid, 3 μl of lipofectamin 2000 CD in 90 μl of Optipro. About 1 μl of injection mix was injected in the dorsal aorta of 2.5-day-old embryos. After the injection, eggs were sealed and incubated until hatching. Hatchlings were grown for 6 weeks until they reached sexual maturity. Semen from males was collected using a female teaser and the massage technique as described in *Chełmońska et al., 2008*. The genomic DNA from semen was extracted and PCR was performed to test for the presence of the transgene in semen. Males showing a positive band in semen DNA were crossed with wild type females. Offspring were

selected directly after hatching by PCR genotyping 5 days after hatching by plucking a feather. For the 16xTF-VNP birds, the CrystallGFP expression was also used for easy screening of transgenic birds. This study was performed in strict accordance with the recommendations in the Guide for the Care and Use of Laboratory Animals of the Monash University. All of the animals were handled according to approved institutional animal care and use committee of Monash University (Research Ethics & Compliance numbers: ERM#27128 and ERM#18809).

## Whole-mount and sections immunochemistry and confocal analyses,and statistics

Embryos were dissected under a Leica fluorescent stereomicroscope and fixed up to 1 hr in 4% formaldehyde at RT. In the case that native fluorescence was required (i.e. no antibody staining), embryos were washed in PBS, incubated in 80% glycerol and directly (i.e. less than 24 hr after fixation) examined. To image native fluorescence expression, the detection spectrum for each fluorophore was set, using a Leica sp5 confocal microscope. The imaging parameters were set according to the sample with the strongest fluorescence to avoid over expression, and sample bleaching. These acquisition parameters were then left the same for the rest of the samples in the same experiment. For section preparations, embryos were embedded in 15% sucrose/7.5% gelatine/PBS solution and sectioned with Leica cryostat at 20 µm. Antibody staining was performed as described in *Serralbo and Marcelle, 2014*. The following primary antibodies were used: anti-GFP chicken polyclonal (ab13970, Abcam;1/1000), anti-Pax7 IgG1 mouse monoclonal (Hybridoma Bank; 1/10), anti-Myosin heavy Chain (MF20) IgG2b (Hybridoma bank; 1/10), anti-Neurofilament IgGIIa (Invitrogen; 1/400), Acetylated Tubulin IgGIIb (T6793, Sigma; 1/500), and anti-HNK1 IgGM (Hybridoma bank; 1/10). Images of native fluorescence and immunostained sections were acquired with a Leica SP5 confocal microscope and an UV-corrected HCX PL APO CS 40x/NA 1.25 Oil immersion objective (WD 0.1 mm), combined with tile scan acquisition. For whole-mount samples, a CARL ZEISS LSM 980 Airyscan 2 confocal microscope on inverted Axio Observer stand was used, with UV-IR corrected PL APO 40x/1.3 oil immersion objective (WD 0.20 mm). Images of native reporter activity in E9 embryos were taken under a Leica 3D Fluorescent microscope with a 4× dry objective.

Mann–Whitney two-tailed non-parametric tests were applied on the entire population of counted cells to evaluate significance of each treatment. **p-value 0.001–0.01; * p-value ≤ 0.05.

## 3DISCO clearing

Performed as described in *Belle et al., 2017*. In short, embryos were dissected under a Leica fluorescent stereomicroscope and fixed for 1 hr in 4% formaldehyde at RT. The embryos were immunostained as described above. Following immunostaining, embryos were dehydrated by immersion in 50, 70, 80, and 100% tetrahydrofurane (THF; in milli-Q water). After dehydration the embryos were rinsed in dichlormethane (DCM) and finally in dibenzyl ether (DBE) to match the refractive index of tissue and surrounding medium leading to transparent sample. Images were acquired on LaVision BioTec UltraMicroscope II based on upright Olympus MVX10 macro zoom microscope with MVPLAPO 2 XC objective (NA 0.50, WD 20 mm) with 2× optical zoom. Overall stack thickness is 3 µm.

## Time-lapse imaging

For time-lapse imaging, embryos were incubated and cultured in custom-made egg incubator described in *Serralbo et al., 2020*, QuailNet database http://quailnet.geneticsandbioinformatics.eu/. Embryos were imaged using Leica SP8 confocal microscope on upright DM6000 stand with HCX APO L 20x/1.00 water dipping (WD 2.00 mm). Z-stack images were taken every 15 min for 9 hr (*Video 3*). Wider field of view time-lapse imaging were acquired every 10 min for 9 hr (*Video 4*) using Leica Thunder Image Model Organism. Images stitched together using the ImageJ software with drift correction plugin.

## Acknowledgements

The authors thank Taryn Guinan from Leica Biosystem for her supervision on Thunder Image Model Organism stereo microscope. The Australian Regenerative Medicine Institute is supported by grants from the State Government of Victoria and the Australian Government. HBT was supported by grants from Stem Cell Australia and the Agence Nationale de la Recherche. We thank the Faculty of Medicine and Health Science for their financial support.

## Additional information

### Funding

No external funding was received for this work.

### Author contributions

Hila Barzilai-Tutsch, Conceptualization, Investigation, Visualization, Methodology; Valerie Morin, Investigation, Visualization; Gauthier Toulouse, Investigation; Oleksandr Chernyavskiy, Stephen Firth, Resources, Visualization; Christophe Marcelle, Conceptualization, Resources, Software, Supervision, Funding acquisition, Writing - original draft, Writing - review and editing; Olivier Serralbo, Conceptualization, Resources, Supervision, Investigation, Visualization, Methodology

### Author ORCIDs

Hila Barzilai-Tutsch ⬛ http://orcid.org/0000-0002-1387-6031
Christophe Marcelle ⬛ http://orcid.org/0000-0002-9612-7609
Olivier Serralbo ⬛ http://orcid.org/0000-0003-0808-3464

### Ethics

This study was performed in strict accordance with the recommendations in the Guide for the Care and Use of Laboratory Animals of the Monash University. All of the animals were handled according to approved institutional animal care and use committee of Monash University (Research Ethics & Compliance numbers: ERM#27128 and ERM#18809).

### Decision letter and Author response

Decision letter https://doi.org/10.7554/eLife.72098.sa1
Author response https://doi.org/10.7554/eLife.72098.sa2

## Additional files

### Supplementary files

• Transparent reporting form

### Data availability

Figure 1 - figure supplement 1-Source Data 1 and Figure 1 - figure supplement 2-Source Data 1 contain the numerical data used to generate the figures.

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
