## [Editor Report]

The manuscript describes several optimizations of classic DNA reporter constructs to monitor closely the dynamics of Wnt/β-catenin signalling during development using transgenic avian lines. As Wnt signalling pathway is essential in the homeostasis of vertebrate and invertebrate organisms, a robust tool to analyse finely the dynamics of the Wnt/β-catenin pathway is of broad interest to biology/biomedicine scientific communities.

---

## [Decision Letter]

**Decision letter after peer review:**

Thank you for submitting your article "Transgenic quails reveal dynamic TCF/β-catenin signaling during avian embryonic development" for consideration by *eLife*. Your article has been reviewed by 3 peer reviewers, and the evaluation has been overseen by Marianne Bronner as the Senior Editor and Reviewing Editor. The following individual involved in review of your submission has agreed to reveal their identity: Aixa Victoria Morales (Reviewer #1).

Essential revisions:

The reviewers found agree that your paper presents a powerful and improved tool to follow Wnt signalling activation in vivo. However, the consensus is that the paper requires extensive revision to demonstrate that the tool is reliable and robust tool. In particular, more transgenic lines are required to demonstrate that this offers a dynamic view of Wnt activation, close to endogenous dynamics. Essential changes are summarized below and elaborated upon in the individual reviews.

1. The authors need to better demonstrate the positive effects of the regulatory elements used in the transgenes. Multiple independent transgenic lines are needed in order to draw firm conclusions. In addition, a demonstration of transgene response to small molecules is needed.

2. Additional experiments should be added to quantitate the expression levels in vivo. The lifetimes need to be better characterized by analysis of kinetic effects of induction and inhibition of b-catenin signalling (e.g. using small molecule agents).

3. As the quail lines may not be generally accessible to many researchers, the authors should provide more data on the expression profiles at different stages of development and better characterize the constructs for use in electroporation.

*Reviewer #1 (Recommendations for the authors):*

In spite of the robustness and usefulness of the new Wnt/β-catenin reporter lines, there are some concerns about the results presented:

1. The proof of the fine temporal resolution of the dynamics of Wnt signalling using the best of the two constructs (16xTCF-VNP) does not seem strong enough. It is mostly based in a 9 hours recording (Sup. Video 3) of somitic cells. It is unclear how long does it take to the two somitic cells indicated by magenta arrow to change their EGFP levels and also it is unclear if they are more intense at the end of the video because they were out of focus during the first frames of the video. It would be more convenient to explore the property of temporal resolution of 16xTF-VNP in a highly dynamic context such as the segmentation clock in presomitic mesoderm at E2 (similar to Video 4). In that tissue, cyclic expression with a periodicity of around 90 min (chicken) or 120-140 min (mouse) has been shown for several components of the Notch and Wnt signalling pathway ( Axin2-T2A-VenusPEST reporter mouse line in Sonnen et al., Aulehla lab; Cell, 2018). In fact, this recent important work has not been acknowledged by the authors.

2. In relation to Supplementary Video 4 (embryos stage E2), the authors speculate that "the reporter-positive cells were found scattered in a wide region of early limb ectoderm, intermingled with reporter negative cells. As the limbs grew, reporter-positive cells migrated towards and coalesced to form the AER, suggesting that they constitute a population of AER progenitors. ". In fact in sections of embryos stage E3, they show how 16xTCF-VNP drives EGFP expression in limb bud mesoderm (besides the expression of AER). In the video at E2 it is unclear if the migrating cells in the hindlimb bud are mesodermal or ectodermal cells. Authors should clarify these aspects (showing E2 sections to demonstrate reporter GFP expression in ectodermal cells before AER formation) or at least, tone down their interpretations.

*Reviewer #2 (Recommendations for the authors):*

Some data are presented to suggest that the 16x multimerised line is more sensitive than the 12x multimerised line based on the expression level in neural crest cells in transient expression experiments and in somitic tissue in the transgenic lines. These data are suggestive but not very quantitative. Can this be improved, for instance could embryos/tissues be dissociated and fluorescence intensity be measured in flow cytometry to quantify these effects, preferably at more than one time point?

It is also suggested that the 16x reporter is more dynamic since it drives expression of a destabilised GFP variant, however there are no direct measurements of lifetimes to support this. It would be good to try to measure this and demonstrate that this is indeed the case, for instance by treating embryos with a small molecule Wnt/β catenin inhibitor and try to measure the dynamics in the response in both strains. It would also be very useful to have an estimate of the time it takes for the reporter to come on, for instance by locally applying a signal and measuring the kinetics of the response. It is somewhat worrying that in the transfection experiments this appears to take a very long time (up to 24 hours?).

It would be really useful to have a stable as well as destabilised reporter and it would be useful to know what these half times are in-vivo and whether they vary development.

The 16x constructs is quite complex, it contains domains known to enhance translation in *Drosophila*, but it is not clear from the data provided that these help or are necessary in the chick. If there is any information on this it would be helpful to provide this, to guide sensible decisions on future construct complexity.

An initial characterisation of the reporters showed that at stages E3 and E9 of development activity is seen in places where it is expected. It would however be good to see what the activity looks like at some earlier stages n development (gastrulation/neurulation) when Wnt signalling is known to play a critical role in patterning and differentiation. Since the embryo geometry is simpler at these stages, it will also be easier to document its activity, both in fixed and live preparations. Furthermore since in early development changes in signalling occur fast this will is will also provide better insight in the sensitivity and kinetics of the reporters.

*Reviewer #3 (Recommendations for the authors):*

For Figure 1, 2 and 3, the figure legends state that anti-GFP antibodies were used on the embryos and sections before imaging. Does this mean that the endogenous fluorescence was not of sufficient intensity to survive fixation and cryopreservation? What do the fluorescent embryos appear under a stereomicroscope? Can the embryos be live imaged during embryogenesis? The Supplementary Video 3 of a somite is of interest. Could more tissues be shown?

Other recommendations:

1. Continuation of the regulatory element analysis using electorporation in the neural tube.

2. If possible, create independent transgenic lines.

---

## [Author Response]

Essential revisions:The reviewers found agree that your paper presents a powerful and improved tool to follow Wnt signalling activation in vivo. However, the consensus is that the paper requires extensive revision to demonstrate that the tool is reliable and robust tool. In particular, more transgenic lines are required to demonstrate that this offers a dynamic view of Wnt activation, close to endogenous dynamics. Essential changes are summarized below and elaborated upon in the individual reviews.1. The authors need to better demonstrate the positive effects of the regulatory elements used in the transgenes. Multiple independent transgenic lines are needed in order to draw firm conclusions. In addition, a demonstration of transgene response to small molecules is needed.2. Additional experiments should be added to quantitate the expression levels in vivo. The lifetimes need to be better characterized by analysis of kinetic effects of induction and inhibition of b-catenin signalling (e.g. using small molecule agents).3. As the quail lines may not be generally accessible to many researchers, the authors should provide more data on the expression profiles at different stages of development and better characterize the constructs for use in electroporation.

Many questions were raised by reviewers on the reliability of the reporters we have used (12XTFd2EGFP and 16xTF-VNP). As explained in the manuscript, the TCF/LEF binding sites (TCF BS) are well-established tools which have been used for decades in literally hundreds of publications to monitor canonical Wnt signaling, starting with the seminal work of Korinek et al., in 1997. Such constructs were shown to efficiently respond to various members of the pathway, in vivo and in vitro

(http://web.stanford.edu/group/nusselab/cgi-bin/wnt/inhibitors; http://web.stanford.edu/group/nusselab/cgi-bin/wnt/activators_detectors), as well as to small activating and inhibiting pharmacological compounds (https://web.stanford.edu/group/nusselab/cgi-bin/wnt/smallmolecules). Altogether, there is a wealth of data showing that TCF BS reliably respond to Wnt signaling.

Twelve years ago (Rios et al., 2010), we have built and extensively characterized the 12XTFd2GFP Wnt reporter used here and compared it to existing constructs containing 3 or 8 TCF-BS.

We then showed that an increase in the number of TCF-BS led to a significant enhancement of sensitivity to Wnt signaling that was not detrimental to the reporter accuracy, since the 12XTFd2GFP responded positively to an activated form of b-catenin (Rios et al., 2010) and was strongly repressed by the Wnt-inhibiting molecule Dkk1 (Sieiro et al., 2016). As the point of this paper was to combine the Wnt reporter to a destabilized fluorophore, we showed that the expression of the destabilized GFP reporter protein was similar to its mRNA, suggesting that 12XTFd2GFP provides a precise view of cells actively engaged in Wnt signaling in vivo. Those important elements have been added to the text (page3, last §) for more clarity.

The 16xTF-VNP construct builds on the same logic of multimerizing the TCF-BS (up to 16) and therefore it should bear the same reliability to Wnt signaling. An important additional point is the use of translation enhancers, which we show are effective in vertebrates (to a point that we now routinely use them in many constructs we generate in the lab). The point of the destabilized fluorophore is discussed below (response 1 to reviewer 1). Even though a multimer of 16 TCF-BS had already been successfully used in *Drosophila* (in Perrimon’s lab; Das Gupta et al., 2005), we were concerned that the basal level of 16xTF-VNP activity may be too high to be useful in our system. In the previous version of the manuscript, we showed that the fluorescent signal of 16xTFVNP was strongly repressed by a dominant form of LEF1 (DN LEF). As using a dominant negative form of LEF1 to repress a TCF/LEF binding site may not be the best of all experiments, we now performed an additional experiment showing that 16xTF-VNP fluorescence is also robustly repressed by the Wnt signaling repressor Axin2 (Sup Figure 1K-N). We believe these explanations and additional data should satisfy the reviewers’ concerns on reliability.

To further characterize the effect of the use of regulatory elements to enhance the activity of the reporter, we have performed additional electroporation experiments that allow quantification of the signals obtained in vivo with (i) the 12XTFd2GFP; (ii) a 16xTF-VNP construct without the translation enhancers (this construct was not used in the previous version of the manuscript) and (iii) a 16xTF-VNP construct with the translation enhancers. These experiments are shown in Sup Figure 2. In addition, additional pictures of side-by-side embryos at the same developmental stage (Figure 4, stage HH13: about 20 somites) show that the fluorescent signal is very similar in the two lines, but significantly stronger in the 16xTF-VNP than in the 12XTFd2GFP. This observation also supports the premise that there is no significant positional effect of the transgene insertion into the genome.

To address the request for more developmental stages, we now provide 3 novel main figures and 2 Sup Figures that illustrate the expression of the reporters in embryos at E2+ in whole mount (Figure 4) and on sections (Figure 5), at E4 in whole mount (Figure 6), during gastrulation (E1, Sup Figure 7) and at E2- (Sup Figure 9).

Reviewer #1 (Recommendations for the authors):In spite of the robustness and usefulness of the new Wnt/β-catenin reporter lines, there are some concerns about the results presented:1. The proof of the fine temporal resolution of the dynamics of Wnt signalling using the best of the two constructs (16xTCF-VNP) does not seem strong enough. It is mostly based in a 9 hours recording (Sup. Video 3) of somitic cells. It is unclear how long does it take to the two somitic cells indicated by magenta arrow to change their EGFP levels and also it is unclear if they are more intense at the end of the video because they were out of focus during the first frames of the video. It would be more convenient to explore the property of temporal resolution of 16xTF-VNP in a highly dynamic context such as the segmentation clock in presomitic mesoderm at E2 (similar to Video 4).

The tool we have designed is appropriate to explore dynamic events that have a velocity compatible with the half-life of the Venus-PEST, which is about 90-110 minutes (see below). The changes of fluorescence we observed in Sup video 3 are true changes in Wnt activity. We have now added a timeline to this video, which shows that the increase in activity (from lowest to highest) takes about 5 hours. To address a related comment from this reviewer (“(cells) were out of focus during the first frames of the video), as mentioned in the manuscript, this video was taken with a confocal microscope, which means that all cells are in focus along the entire video. Regarding the segmentation clock, it cycles in chicken PSM with a periodicity of 90’, which we believe is too short to be detected with this tool. Accordingly, in the new Figure 4 and Sup Figure 9, we do not see any sign of periodicity of the reporter fluorescence in the PSM. The paper of Sonnen et al., that is mentioned by reviewer 1 is a KI of Venus-Pest in the axin2 locus. While the Venus-Pest they used should have the same half-life than ours, the periodicity of axin2 expression in mouse was estimated in their work to be 144 minutes, i.e. more compatible with the half-life of VenusPEST.

To draw a parallel with the design of Notch reporters that allowed the detection of cyclic Notch signaling in the PSM, Auhlela et al., Delaune et al., Masamizu et al., Sorolodoni et al., resorted to various tricks to further destabilize the fluorescent reporter they used, for instance by adding RNA destabilizing sequences in the 3' UTR. A similar strategy was used by Shimojo et al., to detect cyclic variations of Notch signaling in neurons. The transgenic animals that were used in these studies were designed specifically for that purpose and because the signals they generate is so weak, one can see in the literature that these lines are very seldom used.

As mentioned in the manuscript, the signal of a reporter is linearly linked to its stability, and further shortening its half-life will significantly reduce the signal. We have chosen a compromise between high signal and short half-life that may not be suitable for all purposes. We have mentioned the limitation of the reporter lines we designed to detect dynamic changes in Wnt signaling with a short periodicity (2nd§ page 9).

2. In relation to Supplementary Video 4 (embryos stage E2) , the authors speculate that "the reporter-positive cells were found scattered in a wide region of early limb ectoderm, intermingled with reporter negative cells. As the limbs grew, reporter-positive cells migrated towards and coalesced to form the AER, suggesting that they constitute a population of AER progenitors. ". In fact in sections of embryos stage E3, they show how 16xTCF-VNP drives EGFP expression in limb bud mesoderm (besides the expression of AER). In the video at E2 it is unclear if the migrating cells in the hindlimb bud are mesodermal or ectodermal cells. Authors should clarify these aspects (showing E2 sections to demonstrate reporter GFP expression in ectodermal cells before AER formation) or at least, tone down their interpretations.

We now provide sections at different AP levels of a stage HH14 16xTF-VNP embryo (Figure 5). One clearly sees that there are TF-positive cells in the ectoderm in the region of the future limb (see panels I-L). These are likely the same AER progenitors that are observed in Sup Video 4.

Reviewer #2 (Recommendations for the authors):Some data are presented to suggest that the 16x multimerised line is more sensitive than the 12x multimerised line based on the expression level in neural crest cells in transient expression experiments and in somitic tissue in the transgenic lines. These data are suggestive but not very quantitative.

We have further quantified the fluorescence that was observed with the 12x and the 16x constructs. To do this, we electroporated the neural tube with 3 constructs: (i) the 12XTFd2GFP; (ii) a 16xTF-VNP construct without the translation enhancers (this construct was not used in the previous version of the manuscript) and (iii) a 16xTF-VNP construct with the translation enhancers.

The results are shown in Sup Figure 2. They show that the change from 12 TCF/LEF1 binding sites (BS; Sup Figure 2 A-C) to 16 BS (Sup Figure 2 D-F), together with a fluorophore change (from a cytoplasmic d2EGFP to a nuclear form of the brighter Venus, destabilized by a PEST sequence), led to a strong increase in the observed fluorescent signal (by about 11 x). The addition of the translation enhancers (Sup Figure 2 G-I) further increased the fluorescent signal by about 60%. Altogether, these data demonstrate the utility of the strategy we followed to improve the TOPflash reporter and it shows that translation enhancers which were developed in invertebrates are also active in vertebrates.

It is also suggested that the 16x reporter is more dynamic since it drives expression of a destabilised GFP variant, however there are no direct measurements of lifetimes to support this.

The half-life of the d2EGFP and the Venus-NLS-PEST are only slightly different. The d2EGFP was designed by Clontech years ago and its half-life is 2 hours. The half-life of Venus-NLS-PEST is reported to be 1.8 hours (Abranches et al., 2013). Years ago, we tested d2EGFP and Venus-NLSPEST half-life (HL) and confirmed that d2EGFP HL is indeed 2 hours, while Venus-NLS-PEST HL was in our hands more towards 90' (unpublished). Venus is faster-folding than EGFP (17.6' versus 25'; ref fpbase.org), which should slightly decrease the reporter response time. All in all, those differences are rather limited and they should not fundamentally change their response to Wnt signaling.

It would also be very useful to have an estimate of the time it takes for the reporter to come on, for instance by locally applying a signal and measuring the kinetics of the response. It is somewhat worrying that in the transfection experiments this appears to take a very long time (up to 24 hours?)

We have extensively characterized the 12XTFd2GFP construct years ago, showing that the EGFP fluorescence is observed in cells that activate the Wnt response (Rios et al., 2010; Sieiro et al., 2016). In the Rios paper, we also compared a stable and a destabilized TF reporter, demonstrating the importance of the latter to obtain a snapshot of the cells actively responding to Wnt. The 16xTF-VNP construct is a variant of the 12x and it reacts similarly. Even though there is likely a delay between the binding of Wnt to its receptor and an observable fluorescence, it is also the case for a transcriptional response to any signaling pathway. Reviewer 2 seems surprised about the 24 hours delay between electroporation and observation: this delay was just out of convenience (electroporating one day, observing the next day, at a time that we know Wnt signaling is still active), but not necessary.

The 16x constructs is quite complex, it contains domains known to enhance translation in *Drosophila*, but it is not clear from the data provided that these help or are necessary in the chick.

See above response 1 to the same reviewer 2. Note that we have added a note about an additional IVS cassette upstream of Syn21 in the text (last § page 4).

An initial characterisation of the reporters showed that at stages E3 and E9 of development activity is seen in places where it is expected. It would however be good to see what the activity looks like at some earlier stages n development (gastrulation/neurulation) when Wnt signalling is known to play a critical role in patterning and differentiation.

We now provide a picture (Sup Figure 7) that shows the fluorescence in the posterior end of a gastrulating 16xTF-VNP embryo.

Reviewer #3 (Recommendations for the authors):For Figure 1, 2 and 3, the figure legends state that anti-GFP antibodies were used on the embryos and sections before imaging. Does this mean that the endogenous fluorescence was not of sufficient intensity to survive fixation and cryopreservation? What do the fluorescent embryos appear under a stereomicroscope? Can the embryos be live imaged during embryogenesis? The Supplementary Video 3 of a somite is of interest. Could more tissues be shown?

GFP and its variant Venus do not like fixation and their fluorescence tends to fade quite quickly. In contrast, RFP and variants are resistant to fixation. To observe native GFP/Venus after fixation, we image embryos very fast, typically the next day after a step of glycerol-mediated clarification. This point has been clarified in the Materials and methods section (page 10, 2nd §). When we need to apply additional antibodies, for instance in Figures 1, 2 and 3, those experiments take days, and we need to restore the GFP/Venus signal with GFP-specific antibodies. While GFP antibody staining can enhance faint signals, it also tends to level down subtle changes in fluorescence intensity. Evidently, those lines were created to be able to perform live imaging, such as those shown in videos 3 and 4 and it is likely that there will be more to come from our and other labs when the new TOPflash line will be up and running.